# Pine Wilt Disease Segmentation with Deep Metric Learning Species Classification for Early-Stage Disease and Potential False Positive Identification

Nikhil Thapa [1], Ridip Khanal [1,2], Bhuwan Bhattarai [3] and Joonwhoan Lee [1,*]

[1] Department of Computer Science and Engineering, Jeonbuk National University, Jeonju-City 54896, Republic of Korea; nikhiltesla@gmail.com (N.T.); ridipk@gmail.com (R.K.)

[2] Department of Computer Science and Applications, Mechi Multiple Campus, Tribhuvan University, Bhadrapur 57200, Nepal

[3] Department of Digital Technology, Madan Bhandari University of Science and Technology, Thaha Municipality-9, Makwanpur 44100, Nepal; bhubon240@gmail.com

[*] Correspondence: chlee@jbnu.ac.kr

**Abstract:** Pine Wilt Disease poses a significant global threat to forests, necessitating swift detection methods. Conventional approaches are resource-intensive but utilizing deep learning on ortho-mapped images obtained from Unmanned Aerial Vehicles offers cost-effective and scalable solutions. This study presents a novel method for Pine Wilt Disease detection and classification using YOLOv8 for segmenting diseased areas, followed by cropping the diseased regions from the original image and applying Deep Metric Learning for classification. We trained a ResNet50 model using semi-hard triplet loss to obtain embeddings, and subsequently trained a Random Forest classifier tasked with identifying tree species and distinguishing false positives. Segmentation was favored over object detection due to its ability to provide pixel-level information, enabling the flexible extension of subsequent bounding boxes. Deep Metric Learning-based classification after segmentation was chosen for its effectiveness in handling visually similar images. The results indicate a mean Intersection over Union of 83.12% for segmentation, with classification accuracies of 98.7% and 90.7% on the validation and test sets, respectively.

**Keywords:** pine wilt disease detection; segmentation; deep metric learning; forest resource monitoring; disease classification





## 1. Introduction

Pine Wilt Disease (PWD) is caused by the pinewood nematode *Bursaphelenchus xylophilus*, which affects trees in the Pinaceae family. The disease results in a significant decrease in resin secretion, a gradual decline in needle transpiration, and the yellowing and reddening of needles. *B. xylophilus* originated in North America but has since spread to other countries, causing significant environmental damage, as PWD-affected trees ultimately die without human intervention. In the two decades following its introduction to China, tens of thousands of acres of forest were affected due to delayed detection and intervention. More than fifty countries have classified PWD as an imported quarantine disease due to its rapid spread and associated risks. Early detection of PWD is crucial but challenging, as infection becomes severe within five weeks. To manage PWD, quick tracking and containment is essential, which involves the removal of the infected trees through cutting, burning, or chemical and biological methods. While ground surveys have traditionally been used for detection, they require trained professionals familiar with the specific forest of interest, which makes assessments time-consuming, limited in range, and potentially costly depending on the terrain and road conditions [1,2].

To meet the growing need for the efficient and large-scale detection of PWD, AI-based methods have emerged as valuable tools. Remote sensing (RS) stands out as a crucial

technique for forest monitoring, providing extensive coverage, large-scale observation capabilities, and indirect data collection. However, satellite remote sensing, while offering broad-scale monitoring capabilities, faces challenges such as susceptibility to weather conditions, limited temporal resolution, and other factors. As a result, obtaining real-time, high-resolution images becomes challenging, impacting the timeliness of observation data. Additionally, the difficulty in obtaining sufficient samples of specific objects in remote sensing hampers the efficacy of object feature extraction for deep learning networks [3].

Numerous studies have demonstrated the effectiveness of hyperspectral imagery in analyzing PWD with high accuracy [4–9]. However, the use of multispectral cameras presents challenges due to their costliness and higher instability compared to standard RGB cameras. As PWD continues to be a global concern, there is a growing reliance on machine learning technology for its automatic detection. Unmanned Aerial Vehicle (UAV) images have emerged as a primary method for early disease diagnosis. In this approach, a high-quality orthophotograph is typically divided into smaller segments, and a conventional supervised classifier is employed to identify disease locations within a confined area [1].

Researchers have used aerial photographic data from UAVs, using deep learning frameworks and artificial intelligence technology to detect PWD in pine trees. The new deep learning-based methods, primarily employing convolutional neural network (CNN) architectures and segmentation models, have shown promising results in the automatic detection of crop and plant diseases using UAV imagery.

You et al. [1] employed a two-stage detection process using convolutional neural networks and object detection models, achieving high accuracy in both patch-based and real-world tests by localizing the disease and 'disease-like' hard negative samples. Lim and Do [10] utilized SegNet for semantic segmentation and YOLOv2 for object classification in their study on automatic detection of PWD data. Deng et al. [11] utilized UAV images and deep learning, while Han Z et al. [12] proposed the MSSCN model that incorporated spatial attention to detect standing dead trees in different forest environments. Ren et al. [13] introduced a global multi-scale channel adaptation network to improve feature extraction, and Hu et al. [14] proposed a DDYOLOv5 network with Efficient Channel Attention and hybrid dilated convolution modules using UAV images for accurate pine forest disease detection and severity classification. These works have highlighted the growing significance of applying deep learning techniques to address PWD-related challenges.

Among the existing strategies, object detection can identify and locate multiple objects within a single image. This technique has been shown to be effective in detecting plant diseases and pests [15]. Object detection methods typically excel in locating and outlining individual objects in an image, which may be suitable for scenarios where object boundaries are distinct. In the case of PWD identification, the disease does not necessarily manifest in well-defined, isolated regions. It often involves a more subtle and widespread change in the pine tree's foliage and structure. In comparison to object detection, semantic segmentation assigns a class label to every pixel in the image, and thus provides a holistic understanding of the disease's extent along with its impact on the entire tree [16]. Fine-grained analysis is invaluable for assessing the severity of PWD and devising effective mitigation strategies. Also, after object boundaries are drawn post-segmentation, a bounding box can be drawn around the segmented region for further downstream tasks.

However, detecting PWD with UAVs with high accuracy remains a significant challenge. A key issue is potential misidentification due to similar symptoms in other tree species. Additionally, the evolving nature of PWD presents challenges, such as very early-stage infections resembling healthy trees and late-stage infections resembling other natural features. This variability diminishes the efficacy of color-based detection algorithms. Consequently, accurate identification and management of PWD outbreaks using UAV imagery is hindered by these collective challenges [1]. These challenges inevitably lead to the occurrence of false positives, requiring additional correction and verification procedures. Specifically, when accurately classifying the detected instances, features common to diseased regions are inadequate for distinguishing between different species. Furthermore,

since features extracted from non-diseased regions may exhibit some similarities with diseased regions, it becomes crucial to incorporate features that exhibit intra-class dissimilarity to reduce the occurrence of erroneous identifications. Additionally, such features can aid in identifying false positives during the subsequent stage, which are misidentifications occurring during the initial detection. Compared with semantic segmentation, object detection lacks sufficient details about boundary (shape) or size information, which are essential to evaluate the PWD damage, determine the number of infected trees, and to plan the removal of the infected trees.

To address the limitations of object recognition models in detecting PWD, Hwang et al. [17] investigated the accuracy of semantic segmentation, which excels at recognizing objects but typically lacks the ability to precisely capture their location, shape, or area. They evaluated various semantic segmentation models such as SegNet, FCN, U-Net, and DeepLabv3, achieving a maximum accuracy of up to 90%.

Similarly, Xia et al. [18] conducted a study to evaluate the effectiveness of various segmentation models in extracting infected pine trees from UAV images for PWD detection. Their findings underscored the importance of the Atrous Spatial Pyramid Pooling (ASPP) module, which encodes multi-scale contextual information, and an encoder–decoder architecture that helps retain location and spatial details. This combination significantly improved performance when segmenting infected pine trees. This study also highlighted that the choice of backbone architecture plays a crucial role in segmentation accuracy, with ResNet34 and ResNet50 emerging as the most effective backbones. However, increasing the depth of the backbone did not yield significant improvements in segmentation accuracy.

The YOLO (You Only Look Once) family of algorithms offers a different approach, functioning as single-stage target detection models where only one feature extraction step is needed to detect objects. Since its introduction by Joseph Redmon et al. [19] in 2015, the YOLO family has seen several iterations and advancements, including YOLOv1 through YOLOv8, each with unique features and improvements.

YOLOv1 [19] introduced a single-shot detection model with faster and better generalization performance compared to traditional R-CNN-based methods. YOLOv2 [20] added anchor boxes, which are predefined bounding boxes to aid in object localization. YOLOv3 [21] utilized the Darknet-53 backbone, logistic classifiers, and Binary Cross-Entropy (BCE) loss for better object detection. YOLOv4 [22] introduced a Bag of Freebies (BoF) and a Bag of Specials (BoS), offering data augmentation techniques like CutMix, CutOut, and Mixup to boost accuracy without additional inference cost, as well as non-linear activations and skip connections for enhanced performance.

YOLOv5 added a Cross-stage Partial (CSP) connection block to the backbone, improving gradient flow and reducing computational cost. YOLOv6 introduced anchor-free detection and decoupled heads for separate classification and regression tasks. YOLOv7 [23] used an Extended Efficient Layer Aggregation Network (E-ELAN) to enhance training efficiency and feature learning. Finally, YOLOv8 [24], the latest iteration, builds on these advancements with anchor-free detection, a C2f module, a decoupled head, and an improved loss function designed to enhance detection performance and accuracy. With its real-time capabilities and efficient architecture, YOLOv8 is a robust choice for PWD-specific instance segmentation and can effectively support pixel-level detection tasks.

Once the diseased tree has been detected, tree species classification aids in swiftly locating infected regions and optimizing forest management by directing resources to vulnerable species for monitoring and control, guiding species selection for restoration, tailoring silvicultural practices, and informing genetic research for developing resistant varieties, thereby ensuring forest health and sustainability. As the disease features share a close resemblance with healthy trees and other ambiguous objects in the images, a technique specializing in classifying visually similar features is required. Traditional machine learning techniques require feature engineering steps prior to classification or clustering tasks, whereas deep learning directly learns higher-level data representations

within the classification structure. Deep Metric Learning (DML) combines deep learning with metric learning to learn similarity metrics directly from data.

DML involves learning a metric space where semantically similar objects are closer to each other than dissimilar ones. The success of these networks relies on their ability to understand the similarity relationship among samples. DML has been successfully applied to object classification, including scenarios with limited training data, such as few-shot learning. DML excels in addressing inherent challenges in data classification tasks, such as pose variations, illumination differences, scaling, background noise, occlusion, and expression, commonly encountered in problems such as image retrieval, face recognition, and clustering, particularly when traditional distance metrics fall short [25].

In DML, deep neural networks are trained to map input data points into a high-dimensional embedding space, where similar data points are closer to each other and dissimilar data points are farther apart, by optimizing network parameters based on specific loss functions (e.g., contrastive loss or triplet loss). Triplet loss is a loss function in DML, often used for tasks such as image classification and face recognition [26–28], which minimizes the distance between an anchor and a positive sample while maximizing the distance to a negative sample. Various triplet selection strategies, like semi-hard triplet mining, as used on FaceNet [29], aim to strike a balance between excessively challenging and overly simplistic examples, located within a predefined margin hyperparameter. DML has been applied to tasks such as face classification, hyperspectral image classification, and few-shot classifications of diseases [25,29–34], where it has outperformed the baseline methods.

Previous studies have utilized AI and deep learning techniques to identify PWD-affected trees and classify species, encountering challenges with ambiguous objects sharing visual similarities. This paper addresses these challenges by leveraging image segmentation for pixel-level insights in PWD detection, coupled with DML for extracting discriminative features to better distinguish similar images. Specifically, we segment PWD-affected regions from UAV-captured images and mitigate false positives using a subsequent classifier that utilizes DML-extracted features to classify the species of the PWD-affected trees.

To the best of our knowledge, the identification of possible false positives using DML in PWD has not been previously explored. Additionally, there have been no direct comparisons between the performance of instance segmentation and the classification of diseased regions based on cropping bounding boxes, likely because of their fundamentally different approaches. Our study aims to advance PWD identification and management by combining a holistic understanding of the disease's extent and its impact on the PWD-affected trees obtained by semantic segmentation and DML-based features to better classify similar types of images. The key contributions of this research include:

1. Use of the YOLOv8 large-segmentation (YOLOv8l-seg) model for instance segmentation to identify trees affected by PWD, combined with the use of class-wise segmentation to classify tree species, enabling a comparison of two segmentation approaches: a simple two-class model versus a method that first segments by species and then classifies based on disease.
2. Generation of distinctive features for accurate disease classification using DML; the classification model features a Random Forest classification head trained on features extracted from a ResNet-50 backbone which uses semi-hard triplet loss.
3. Addressing the issue of false positives using a dedicated class for false positives in the species classifier. This approach allows for targeted adjustments to distinguish between real positives and incorrect identifications.

## 2. Materials and Methods

### 2.1. Data Acquisition, Dataset Description, and Preprocessing

The dataset consisted of 10,063 images with a resolution of 428 × 428 pixels and a .jpg extension, and corresponding ground truth annotations for infection regions stored as .json files, containing label information for the species of the affected pine tree and the stage of the disease, sourced from the Korea Forestry Promotion Institute (KofPI). These

images were captured by a KD2 Mapper drone equipped with a SONY RX1R Mark II (Sony Corporation, Tokyo, Japan) camera, with the average shooting altitude maintained at 140 m and a flight speed of 60 km/h, and mapping was conducted using Pix4D MAPPER software (https://www.pix4d.com/product/pix4dmapper-photogrammetry-software/).

We utilized instance segmentation, a computer vision method pivotal for identifying and localizing objects at the pixel level within images. To ensure compatibility and consistency, we adopted the widely used MS COCO format for annotations. Subsequently, these annotations underwent conversion into YOLO *.txt file format, a crucial step in preparing the data for training. Each row within the text file corresponds to a distinct object instance detected within the image, facilitating streamlined processing and integration into the YOLO framework. This approach ensures the efficient utilization of annotated data for training the YOLO model, enabling robust object detection and localization capabilities.

Regarding classification, the species labels included '*Pinus densiflora*', '*Pinus koraiensis*', and '*Pinus thunbergii*', abbreviated as 'PD', 'PK', and 'PT', respectively. The apparent false positive class, abbreviated as 'AFP', consists of images of the ground, rocks, man-made objects, and deciduous trees in the forest scenario that might be confused for PWD objects during segmentation. These were the images that were detected as diseased but had no corresponding ground truth label. The training dataset for the classifier consisted of image segments extracted from the original images resized to 224 × 224-pixel dimensions after the addition of a margin of 20 pixels to enhance the contextual information and ensure the robustness of the model.

Within the specified categories, the 'PD' and 'PT' classes exhibit significant visual resemblance, which presents a challenge for differentiation; additionally, the 'PT' category comprised a relatively smaller number of images compared to other categories. The visual intricacy in distinguishing 'PT' from 'PD' trees led to the strategic decision to consolidate these classes into a unified 'PD-PT' category, acknowledging the complexities in species classification. PWD manifests in each pine species in different stages of needle discoloration, corresponding to the severity of infection. These stages were labeled as initial (yellowish-red), middle (greyish-white), and late (leafless). The late (leafless) stage of the disease posed difficulty in classification due to the withering and falling of leaves; thus, the images of this stage were exempted from species classification. Example images from each class of the species classifier are shown in Figure 1.

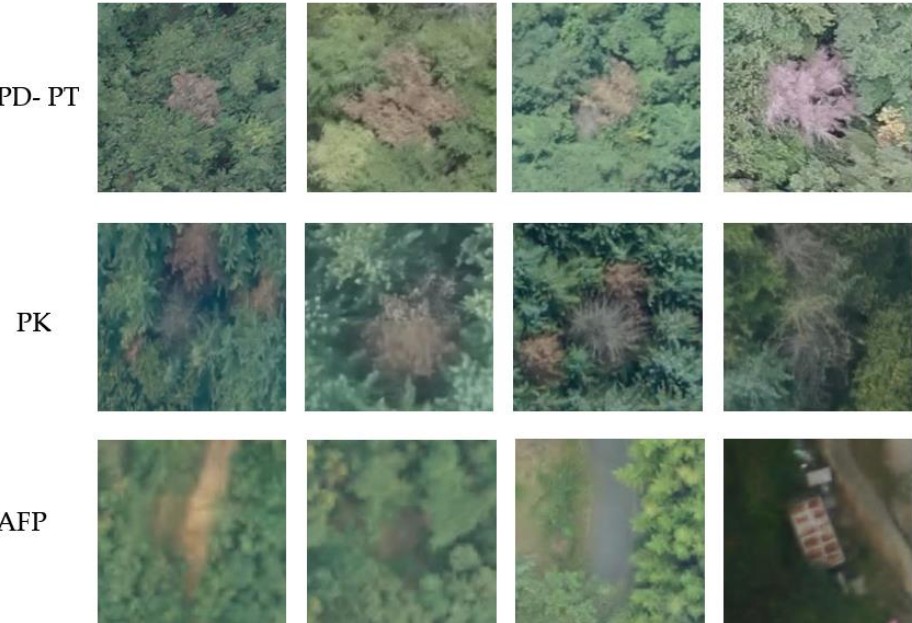

**Figure 1.** Example of images from each class in species classification.

### 2.2. System Overview

As illustrated in Figure 2, this paper presents a novel two-stage framework. Initially, we segmented PWD-infected trees from the background, followed by employing a DML classifier. Our approach utilizes a Random Forest classification head trained on features extracted from a ResNet-50 backbone, leveraging semi-hard triplet loss.

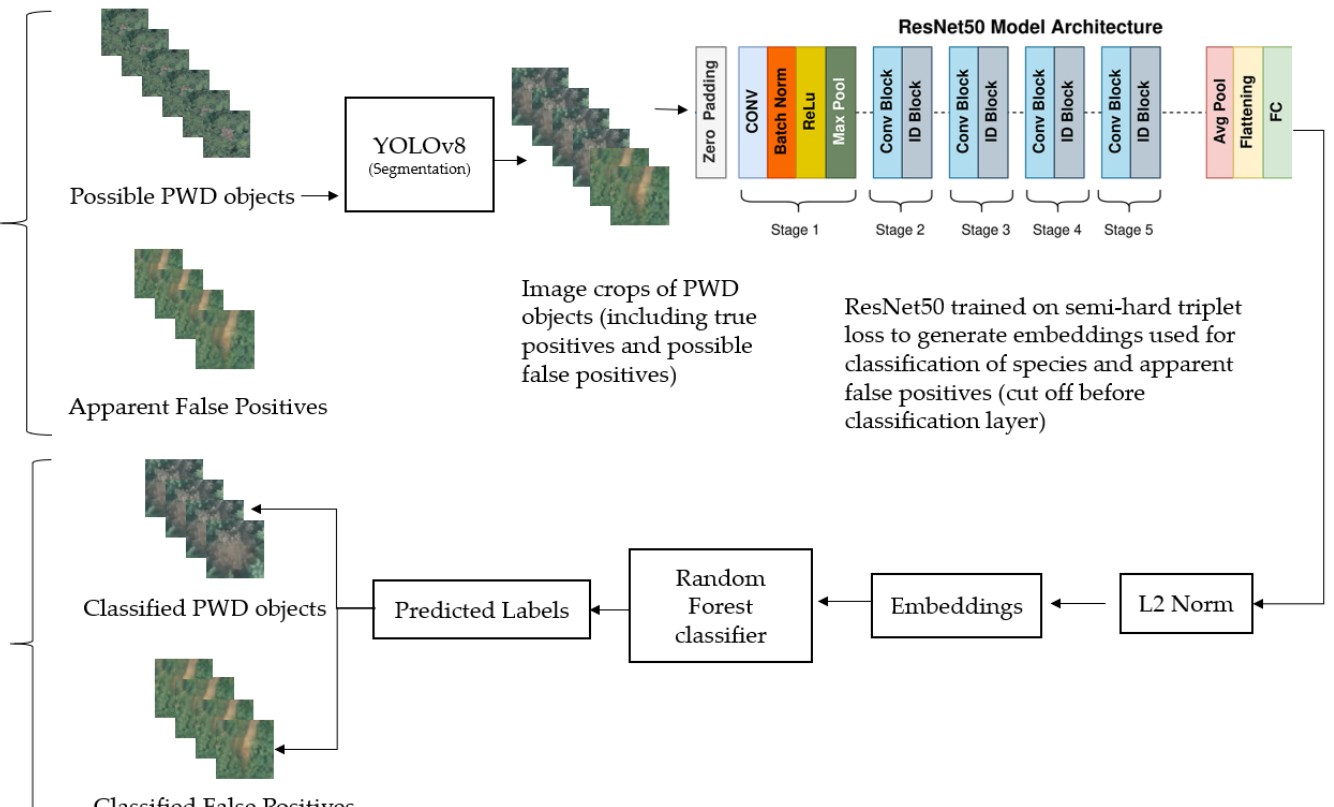

**Figure 2.** Illustration depicting the workflow of the overall system.

In this study, we utilized YOLOv8, a state-of-the-art object detection and segmentation framework, to address the task of segmenting PWD-infected trees from the background. YOLOv8 builds upon the success of its predecessors by incorporating several key improvements aimed at enhancing recognition performance and accuracy.

YOLOv8 is structured into four main modules: Input, Backbone, Neck, and Output. The Input module preprocesses the input data, while the Backbone module extracts hierarchical features from the input image. The Neck module further refines these features, and the Output module generates the final segmentation results.

YOLOv8 utilizes a CSPLayer2Conv (C2f) module within the Backbone and Neck sections, which enhances the model's feature representation capabilities by incorporating additional branches and cross-layer connections. The decoupled detection head adopts an anchor-free approach, which improves localization accuracy and simplifies bounding box regression, thereby facilitating more precise object segmentation.

It combines Binary Cross-Entropy (BCE) loss with regression techniques such as Complete Intersection over Union (CIOU) and Varifocal Loss (VFL). This combination effectively balances the classification and localization objectives, resulting in improved overall segmentation accuracy. Additionally, YOLOv8 employs a task-aligned assigner matching strategy for frame matching. This dynamic approach enhances the model's performance and flexibility, particularly in complex scenarios, by adapting the assignment of tasks based on the input data. In this study, we used YOLOv8l-seg, which denotes the large variant optimized for segmentation tasks.

After obtaining the bounding box coordinates of the detected diseased regions for the test set, taking the largest dimension of the bounding box, a margin of 20 pixels was added, and the resulting dimension was taken as width and height. From the center of the bounding box, this frame was cropped and resized to 224 × 224 pixels. For training and validation, the crops were directly taken from the ground truth annotations. A visual representation of the image crops from segmentation is depicted in Figure 3.

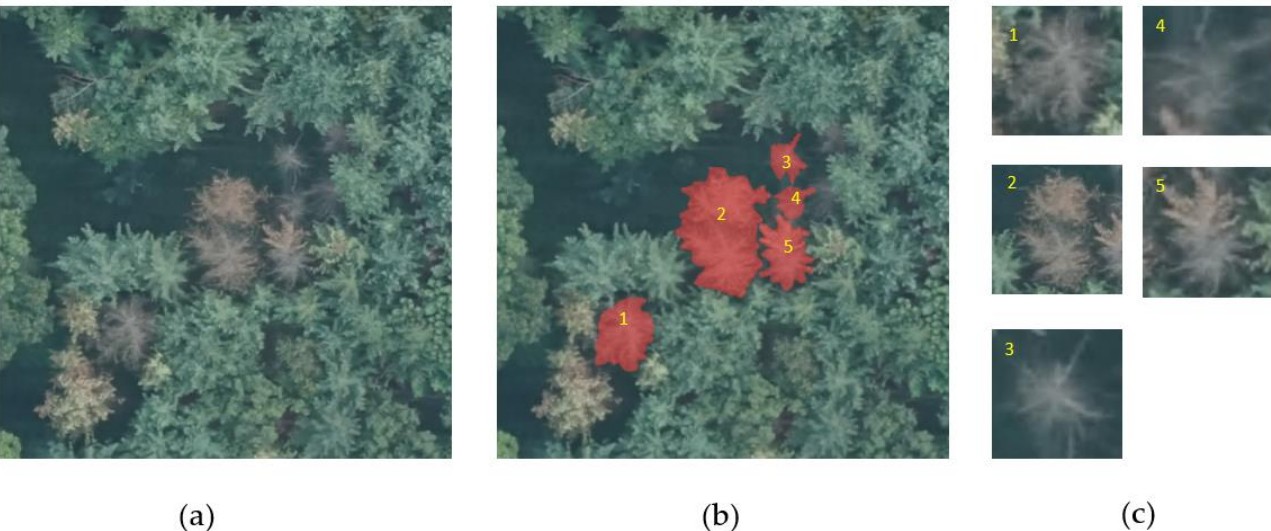

(a)            (b)            (c)

**Figure 3.** Illustration of original image (**a**) and segmented image (**b**), where the numbered segments correspond to the cropped images in (**c**).

For classification purposes, the triplet loss function is a common DML loss function designed to train neural networks to learn such embeddings by pulling together samples of the same class (anchor and positive) while pushing apart samples of different classes (anchor and negative). By optimizing the network with triplet loss, it learns to embed samples in a way that maintains their relative similarities or dissimilarities. Using ResNet50 as the backbone and semi-hard triplet loss as the objective function, feature embeddings were extracted.

For generating the embeddings by implementing semi-hard triplet loss, we initially computed the distances between embeddings, followed by the creation of a binary matrix to detect positive pairs. Subsequently, by inverting this matrix, we effectively segregated the positions for positive and negative pairs, thereby simplifying the process of selecting semi-hard negatives. These semi-hard negatives were identified based on their proximity to anchor points, exceeding the distances of positives. In scenarios where optimal counterparts were not accessible, we designated the farthest negative sample as the alternative, commonly referred to as the hard negative.

Following the generation of the embeddings by the feature-extracting backbone, this information was subsequently channeled into the Random Forest classification head for further processing.

$$\mathcal{L} = \max(d(a, p) - d(a, n) + \alpha, 0) \tag{1}$$

Equation (1) $\mathcal{L}$ represents the semi-hard triplet loss, where $d(a, p)$ and $d(a, n)$ are the distances from the anchor to the positive and negative samples in the embedding space, while the margin $\alpha$ ensures that the distance from the positive to the anchor is greater than the negative distance, and taking the maximum ensures that only the positive differences contribute to the loss. Embeddings thus learned from the ResNet50 backbone are then fed to the Random Forest classifier to train it. To classify the test data, embeddings were generated similarly from the trained model.

*2.3. Experimental Setup*

Experiments were performed using a NVIDIA GeForce RTX 3090 (Nvidia, Santa Clara, CA, USA), with the YOLOv8l-seg model using the pre-trained model. Except for the adjustment of the number of training epochs to 500 and batch size 8, the default hyperparameters were applied for both semantic segmentation and class-wise instance segmentation.

Regarding the classifier, the fastai library [35] was used with a cyclical learning rate [36] and the model was tuned until 95% validation accuracy was achieved. For both the YOLOv8l-seg model and the DML classifier, the training, validation, and testing split ratio was 70:15:15. We also noticed a slight imbalance in the class distribution, as the ratio of images in 'PD+PT' to 'PK' was 10:9. It is to be noted that not all images in the segmentation stage may have diseased objects, while some images may have multiple diseased objects in a single image, thus making the number of datasets in the subsequent classifying stage different from that of segmentation.

## 3. Experimental Results and Discussion

*3.1. Evaluation Metrics*

Precision measures the proportion of correctly predicted positive instances (true positives) out of all predicted positive instances (true positives + false positives). It reflects the model's ability to avoid false positives. Recall calculates the proportion of true positive instances out of all actual positive instances (true positives + false negatives). It indicates how well the model captures all relevant positive samples. Mean Average Precision at IoU 0.5 (mAP50) computes the average precision across different object categories at an IoU (Intersection over Union) threshold of 0.5. It assesses the model's performance in detecting objects accurately, considering a moderate overlap threshold. mAP50-90 (Mean Average Precision from IoU 0.5 to 0.9) extends the evaluation to a range of IoU thresholds from 0.5 to 0.9. It provides a more comprehensive assessment of detection accuracy across varying IoU levels, capturing both strict and lenient matching criteria.

In YOLOv8, the segmentation results combine two important elements: masks and boxes. The 'mask' is essentially a binary representation of an object's segmentation within an image. It identifies which pixels belong to the object (marked as '1') and which do not (marked as '0'). This feature allows YOLOv8 to segment the regions of detected objects accurately, providing a fine-grained understanding of the scene. Such masks are invaluable in a variety of computer vision tasks, including instance segmentation and semantic segmentation, and can also be used in further post-processing applications.

Meanwhile, the "boxes" in YOLOv8 are bounding boxes that tightly enclose detected objects. Each bounding box is defined by four coordinates: the top-left and the bottom-right corners. These coordinates encapsulate the position and size of the object within the image, making bounding boxes essential for localization tasks. The combination of masks and boxes in YOLOv8 offers a comprehensive approach to both object detection and segmentation, providing rich data for computer vision applications.

For the classification task, the metrics used were classification accuracy, precision, recall, and F1 score. Classification accuracy measures the number of images correctly classified and the F1 score is the harmonic mean of precision and recall. Equations (2)–(6) represent the precision, recall, mAP, accuracy, and F1 score, respectively.

$$\text{Precision} = \frac{TP}{TP + FP} \tag{2}$$

$$\text{Recall} = \frac{TP}{TP + FN} \tag{3}$$

$$\text{mAP} = \frac{\sum_{i-1}^{k} AP_i}{k} \tag{4}$$

$$\text{Accuracy} = \frac{TP + TN}{TP + FP + TN + FN} \tag{5}$$

$$F1 = \frac{2 \times \text{Precision} \times \text{Recall}}{\text{Precision} + \text{Recall}} \tag{6}$$

### 3.2. Results and Discussions

In segmentation, the YOLOv8l-seg model achieved 0.966 precision, 0.922 recall, 0.958 mAP40, and a mAP50-95 value of 0.727 for segmentation masks. For bounding box-based results, the model achieved a precision of 0.963. The recall rate of 0.929 is slightly higher than that for segmentation masks. The mAP50 score of 0.962 indicates a high accuracy at the 50% IoU threshold, while the mAP50-95 value of 0.842 suggests the model maintains its accuracy even with varying IoU criteria. The results are summarized in Table 1 and a visual analysis of the segmentation model evaluation indicators are illustrated in Figure 4.

**Table 1.** Results of disease object segmentation (single class).

| Type | Precision | Recall | mAP50 | mAP50-95 |
|---|---|---|---|---|
| disease object (mask) | 0.966 | 0.922 | 0.958 | 0.727 |
| disease object (box) | 0.963 | 0.929 | 0.962 | 0.842 |

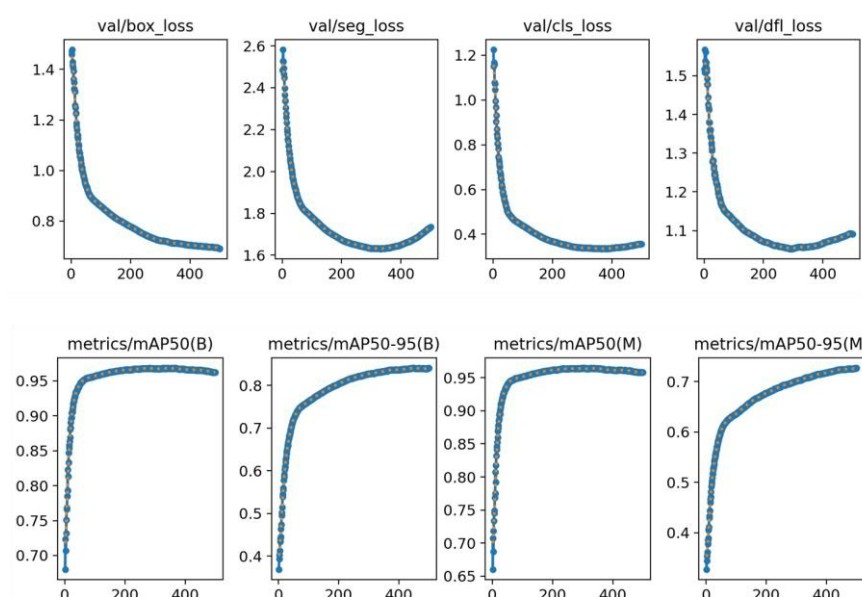

**Figure 4.** Visual analysis of segmentation model evaluation indicators.

The performance of the model was also analyzed for specific classes of objects. For the PD-PT class, the precision for segmentation masks was 0.886, with a recall rate of 0.646. This relatively lower recall rate suggests the presence of some false negatives. The mAP50 score for segmentation masks in this class was 0.787, and the mAP50-95 was 0.561, indicating that the model may require further optimization for improved accuracy and consistency across varying IoU thresholds. For bounding boxes in the PD-PT class, the precision was 0.888, with a recall rate of 0.649. The mAP50 score for bounding boxes was 0.791, and the mAP50-95 value was 0.692.

In the PK class, the precision for segmentation masks was 0.853, with a recall rate of 0.663, suggesting a moderate rate of false negatives. The mAP50 score for segmentation masks in this class was 0.764, while the mAP50-95 value was 0.554. For bounding boxes in the PK class, the precision was 0.855, with a recall rate of 0.669. The mAP50 score for bounding boxes was 0.771, and the mAP50-95 value was 0.662. The results are summarized in Table 2.

**Table 2.** Results of species-wise disease object segmentation (multi-class).

| Type | Precision | Recall | mAP50 | mAP50-95 |
|------|-----------|--------|-------|----------|
| PD-PT (mask) | 0.886 | 0.646 | 0.787 | 0.561 |
| PD-PT (box) | 0.888 | 0.649 | 0.791 | 0.692 |
| PK (mask) | 0.853 | 0.663 | 0.764 | 0.554 |
| PK (box) | 0.855 | 0.669 | 0.771 | 0.662 |

In class-wise segmentation, there are noticeable variations in the model's performance. The PD-PT class shows a lower recall rate for segmentation masks at 0.646, suggesting that some relevant objects are not being captured. The mAP50-95 for PD-PT masks is 0.561, indicating a need for improvements in consistency and accuracy under varied IoU conditions. For bounding boxes in this class, the precision is 0.888, with a recall rate of 0.649, mAP50 of 0.791, and mAP50-95 of 0.692. These values suggest a better performance with bounding boxes compared to segmentation masks. Similarly, in the PK class, the precision for segmentation masks is 0.853, with a recall rate of 0.663, indicating a moderate rate of false negatives. The mAP50-95 for this class is 0.554, the lowest among the metrics, suggesting that the model's consistency could be improved. For bounding boxes, the precision is 0.855, with a recall rate of 0.669, mAP50 of 0.771, and mAP50-95 of 0.662, demonstrating slightly better performance than for segmentation masks but with similar challenges in achieving consistency. Some segmentation results, including the segmented false positives, are shown in Figure 5.

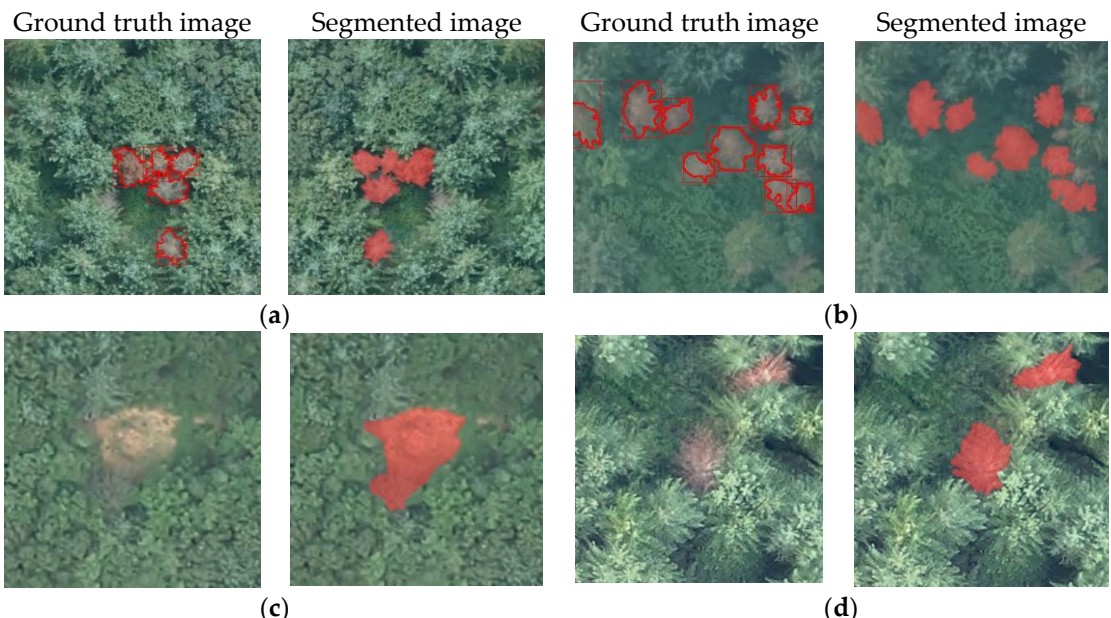

**Figure 5.** Test results of the segmentation model with its ground truth (on the left side) and segmented images (on the right side) with segments as red patches, (**a**,**b**) represent the actual disease objects, whereas (**c**,**d**) represent false positives detected.

The classifier achieved a test accuracy of 90.7%. The confusion matrix is displayed in Figure 6. The weighted average precision was 94%, whereas both recall and F1 score were 91%. The precision, recall, and F1 score achieved on the test set are tabulated in Table 3.

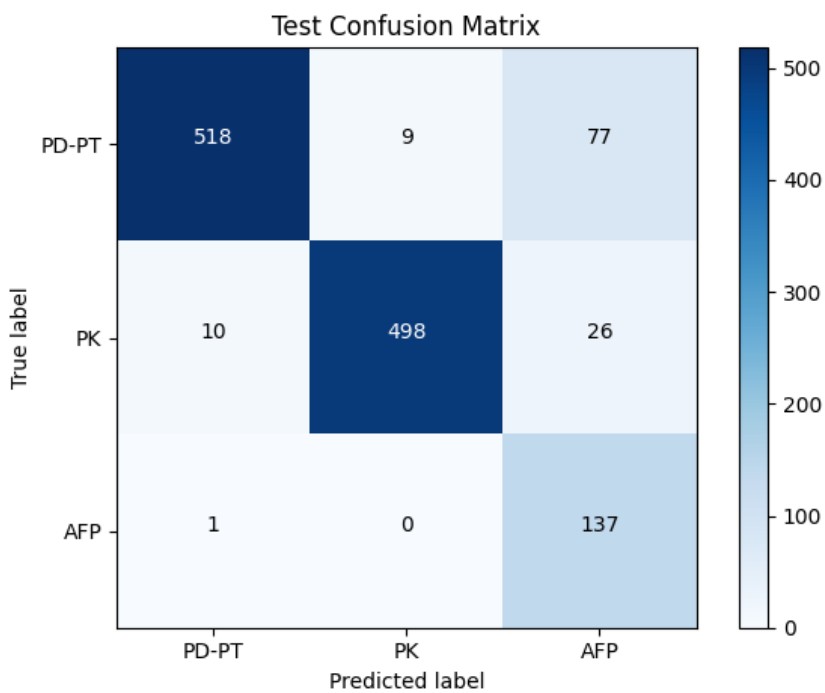

**Figure 6.** Confusion matrix from DML classifier on the test set.

**Table 3.** Class-wise evaluation.

| Classes | Precision | Recall | F1-Score |
|---------|-----------|--------|----------|
| PD-PT | 0.97 | 0.88 | 0.92 |
| PK | 0.98 | 0.92 | 0.95 |
| AFP | 0.59 | 0.99 | 0.74 |

The classification model's results signified that our model excels in distinguishing between the primary classes, namely 'PD-PT' and 'PK', while managing the presence of false positives. The model showcased its proficiency in making nuanced distinctions within a complex and diverse dataset, highlighting its adaptability to the intricacies of real-world data. This study underscores the effectiveness of the DML-based approach to PWD classification.

The precision, recall, and F1 scores for a segmentation model across three distinct classes show varying performance levels. The PD-PT class exhibits a precision of 0.97, recall of 0.88, and F1 score of 0.92, indicating strong accuracy and a good balance between precision and recall. The PK class achieves an even higher precision of 0.98, recall of 0.92, and F1 score of 0.95, demonstrating exceptional performance. However, the AFP class, while having a high recall of 0.99, has a much lower precision of 0.59 and an F1 score of 0.74, suggesting a high rate of false positives.

In our model, the Deep Metric Learning (DML) classifier analyzes cropped images after the initial disease detection, enabling the finer classification of species and filtering out potential false positives by comparing the similarity of generated embeddings. This approach offers a distinct advantage over traditional multi-class segmentation, which relies on feature representations to both detect diseases and classify species simultaneously. By focusing the DML classifier on post-segmentation analysis, the proposed technique achieves enhanced precision and recall while also accommodating additional classifications like AFP.

A key aspect of this method is that it uses similarity-based feature representations to enhance accuracy and identify potential false positives. Integrating the DML classifier into the training loop can further improve the model's precision and recall during the learning process, leading to a more robust segmentation model.

The decision to include a false positive class in the DML-based classifier played a pivotal role, as these occurrences could be rectified. It is worth noting that within the scope of our study, the incidence of false positives was limited, and that their number remained relatively low. Including the false positives as a separate class in the classification task implies that such images could be identified by the model's inference and be rectified or reviewed according to the need for further downstream tasks.

We applied t-SNE visualization to the embeddings generated from the test dataset to gain insights into how well the model's embeddings captured the underlying structure and clustered together, providing a valuable measure of the model's effectiveness in capturing the similarity and dissimilarity of relationships within the data. The t-SNE plots are shown in Figure 7. From the t-SNE visualizations, it was evident that the PD-PT and PK classes in the species classification formed distinct clusters, whereas the false positives exhibited a scattered distribution, even though some outliers exist in the disease classes too.

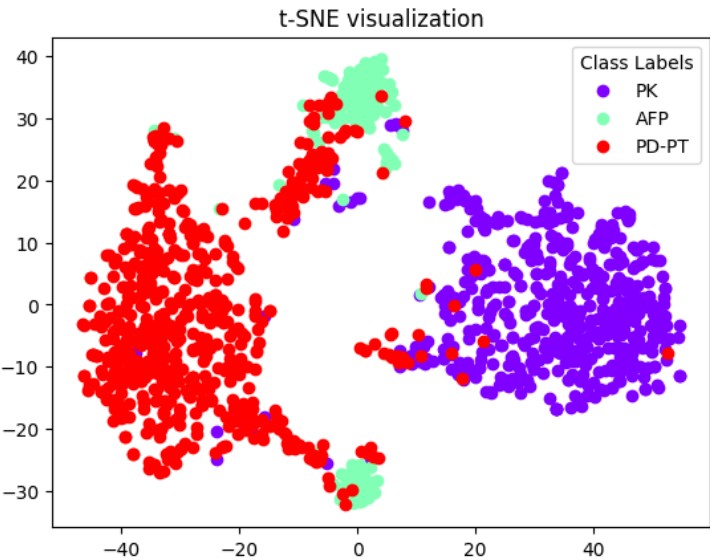

**Figure 7.** t-SNE visualization of the embeddings of the test set.

## 4. Conclusions

PWD poses a significant threat to forest resources, requiring swift detection and intervention. Drone-captured images are useful for detecting PWD efficiently. In this study, we presented a solution for PWD detection through segmentation and classification utilizing the images. We considered a two-stage approach involving disease area segmentation using YOLOv8, followed by classification utilizing a Random Forest Classifier trained on ResNet-50 embeddings obtained through the application of semi-hard triplet loss.

Notably, the YOLOv8l-seg model demonstrated good results on the validation data, achieving a precision of 0.966, recall of 0.922, mAP40 of 0.958, and mAP50-95 of 0.727 for the segmentation mask. Subsequently, the classifier attained a test accuracy rate of 90.7%, with a weighted average precision of 94%, accompanied by the recall and F1 score metrics both peaking at 91%. The model's performance varies when examined by class in segmentation tasks. For the PD-PT class, the recall rate for segmentation masks is only 0.646, indicating that the model might be missing some relevant objects. Additionally, the mAP50-95 score for PD-PT masks, standing at 0.561, suggests room for improvement in maintaining accuracy and consistency across different IoU thresholds.

Future works will focus on developing methodologies specifically designed to effectively mitigate and handle false positives. One potential direction is the exploration of different DML loss functions, such as 'sphereface' [33] or 'AdaCos' [34], which could potentially enhance the model's discriminative power. Approaches that use separate feature sets for object detection and segmentation, and a different similarity-based feature set for

classification, can be effective in self-training and reducing false positives. Techniques like DML allow classifiers to distinguish between different classes and identify potential false positives, providing an added layer of accuracy. The replacement of the YOLOv8 backbone with more specialized architectures designed for aerial segmentation tasks holds promise for improving the model's performance. The incorporation of self-supervised pre-training and the exploration of alternative substitutions for a feature-extracting backbone for DML, fine-tuned to meet the unique demands of the specific task, could contribute to overall performance enhancement.

**Author Contributions:** Conceptualization, J.L.; Formal analysis, N.T., R.K. and B.B.; Funding acquisition, J.L.; Investigation, N.T., R.K. and B.B.; Methodology, J.L.; Project administration, J.L.; Supervision, J.L.; Visualization, N.T.; Writing—original draft, N.T.; Writing—review and editing, N.T. and B.B. All authors have read and agreed to the published version of the manuscript.

**Funding:** This research received no external funding.

**Data Availability Statement:** Data sharing is not applicable to this article.

**Acknowledgments:** We would also like to express our gratitude to the Korea Forestry Promotion Institute (KofPI), the National IT Industry Promotion Agency (NIPA), and the editors of the Writing Center at Jeonbuk National University.

**Conflicts of Interest:** The authors declare no conflicts of interest.

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
