# Peer review of "Pine Wilt Disease Segmentation with Deep Metric Learning Species Classification for Early-Stage Disease and Potential False Positive Identification"

_electronics, doi:10.3390/electronics13101951_

Round 1

Reviewer 1 Report

Comments and Suggestions for Authors

I found the article quite engaging, and I have some constructive feedback to help improve its impact.

   1. Firstly, I think highlighting the novelty of the work would be a great way to capture readers' attention.

  2. Additionally, expanding upon the introduction section and including a table summarizing the critical work would be beneficial to readers. Perhaps a table titled "Table X: A Brief Overview of the Literature" could be added under the introduction to aid in summarizing the work.

In the Experimental
 Results and Discussion section:

       1. Including the ablation study in the table would aid in presenting a more detailed account of the proposed method. It would effectively illustrate the role and significance of each component, leading to a more insightful analysis for the reader.

     2. There may be a small issue with the figure numbering in section 3.1. To ensure clarity, it would be helpful for Figures 3 and 4 to be consecutively numbered. Additionally, it may be beneficial for the authors to review the figure numbering throughout their work to ensure consistency. 

     3. To strengthen the reliability of their proposed segmentation method, it would be beneficial for the authors to conduct a comparative analysis with other state-of-the-art algorithms on the same dataset. This approach can lead to valuable insights and further improvements in the proposed method.

In the Conclusions Section

     The authors should compare their findings with previous studies or state-of-the-art (SOTA) methods in line 384~388.

Other Suggestions:

   To add more value to their work, the authors may consider conducting a comparative analysis of their experimental results with those of related studies based on the same dataset. This could help them highlight the unique contributions and improvements they have made.

Comments on the Quality of English Language

The manuscript requires careful revision of its English description due to numerous grammar and word errors.

Author Response

We greatly appreciate your insightful comments, Attached is our response addressing them. Please see the attachment.

Reviewer 2 Report

Comments and Suggestions for Authors

This study presents a novel method for Pine Wilt Disease detection and classification, where YOLOv8 was used for segmenting diseased areas, followed by cropping the images and applying Deep Metric Learning. Trained a ResNet50 model using Semi-Hard Triplet Loss to obtain embeddings, and subsequently trained a random forest classifier tasked with identifying tree species and distinguishing false positives.

1 There are many "Figure 4" in the article, please modify it

2 The introduction of the data set is not clear enough, such as image type information, image quantity information, etc.?

3、In the method introduction, figure 2 describes the "segment PWD-infected trees from the background", where the segmentation refers to the cut out like in Fig?

4 In the method introduction, we need to explain more clearly how to use YOLOv8 for segmentation and ResNet50 for the generation of embedding vector.

5 Too little experiment content, no comparative experiment?

Comments on the Quality of English Language

Moderate editing of English language required

Author Response

Thank you for the valuable comments, Attached is our response to your comments. Please see the attachment.

Reviewer 3 Report

Comments and Suggestions for Authors

The study presents a method for detecting and classifying Pine Wilt Disease (PWD) using unmanned aerial vehicles (UAVs) images. The approach first segments diseased areas using YOLOv8, followed by image cropping and Deep Metric Learning. Then a ResNet50 model trained with Semi-Hard Triplet Loss is employed to obtain embeddings. A random forest classifier trained on the embeddings is used to classify tree species and false positives.

Please elaborate on the image dataset used, including details such as the original resolution before resizing to 256x256, the number of images in the train/validation/test sets, and the distribution of classes within the dataset.

Line 232: The visual intricacy in distinguishing 'PK' from 'PD' trees … should it be 'distinguishing 'PT' from 'PD''?

Line 234-238: Since the advanced PWD stages pose difficulty in classification due to the withering and falling of leaves, alongside the absence of a standard for discerning the stages and thus were exempted from species classification. Should the title of the paper reflect this? For example, change to 'Pine Wilt Disease Segmentation with Deep Metric Learning Classification for Species with Early Stage Disease and Potential False Positive identification'? The Results and Discussion sections should also address this.

Figure 4: Please display and specify the segmentation results for each class (species and PWD stages).

Please discuss how this approach is superior to instance segmentation models with class PD-PT with PWD, PK with PWD, etc, such models also provide object class label and pixel-level information (the shape). Additionally, instance segmentation can detect and segment multiple objects in one image.

Author Response

Thank you for the valuable comments. We have tried to address the issues, and our response is attached below. Please see the attachment.

Round 2

Reviewer 1 Report

Comments and Suggestions for Authors

Accept in the present form.

Author Response

Dear Reviewer,
We are very thankful for accepting our journal submission. Your insights were invaluable in improving our work. We truly appreciate your time and expertise.

Reviewer 2 Report

Comments and Suggestions for Authors

The authors have replied to all the questions and the modified manuscript was qualified for publication.

Author Response

Dear Reviewer,
Thank you for accepting our journal submission. Your insights were invaluable in improving our work. We truly appreciate your time and expertise.

Reviewer 3 Report

Comments and Suggestions for Authors

Thank you for addressing the comments.

For comment 5, please explain why false positive can not be included as one of the classes for the one-stage instance segmentation model and have class labels consistent with the two-stage model proposed. Besides, what are the benefits of identifying false positive since the study objective is to detect the pine wilt disease accurately?

The one-stage instance segmentation model should be compared and discussed with the results listed in Table 3, as it represents the final output of the two-stage model proposed.
